# A Brief History of *Drosophila* (Female) Meiosis

**DOI:** 10.3390/genes13050775

**Published:** 2022-04-27

**Authors:** Jessica E. Fellmeth, Kim S. McKim

**Affiliations:** 1Department of Biology, Millersville University, Millersville, PA 17551, USA; jessica.fellmeth@millersville.edu; 2Waksman Institute and Department of Genetics, Rutgers, The State University of New Jersey, Piscataway, NJ 08854, USA

**Keywords:** *Drosophila*, oocyte, meiosis

## Abstract

*Drosophila* has been a model system for meiosis since the discovery of nondisjunction. Subsequent studies have determined that crossing over is required for chromosome segregation, and identified proteins required for the pairing of chromosomes, initiating meiotic recombination, producing crossover events, and building a spindle to segregate the chromosomes. With a variety of genetic and cytological tools, *Drosophila* remains a model organism for the study of meiosis. This review focusses on meiosis in females because in male meiosis, the use of chiasmata to link homologous chromosomes has been replaced by a recombination-independent mechanism. *Drosophila* oocytes are also a good model for mammalian meiosis because of biological similarities such as long pauses between meiotic stages and the absence of centrosomes during the meiotic divisions.

## 1. Introduction

Crossing over is required for the segregation of homologous chromosomes at meiosis I. This relationship has been demonstrated in Drosophila females in several ways, such as the increase in nondisjunction when crossing over is reduced [1,2]. Meiosis begins at around the time of oocyte specification, proceeds through the process of recombination, and ends with two rounds of divisions. This review focuses on female meiosis, because like most other organisms, it depends on the initiation and function of recombination. Meiosis in *Drosophila* males, in contrast, is fascinating because it shows it is possible to achieve accurate chromosome segregation with a completely different system of homolog interactions [3]. For more details and a comprehensive list of meiotic genes, the interested reader is referred to an extensive review of meiosis in female Drosophila including a compendium of most of the important genes [4].

Drosophila females are an excellent system to study meiosis because they combine powerful genetics and cell biology. Errors in meiosis can be measured using standard genetic crosses that detect sex chromosome aneuploidy, or FISH to detect the aneuploidy of all chromosomes. Classical mutants or RNAi (using shRNA) are used to knock out or knock down any gene during meiosis. Tissue-specific gene expression systems (UAS/GAL4) allow for germline-specific knockouts and the expression of mutants, which is important when the genes are essential. These tools also allow for short-term expression or knock down experiments, which facilitates investigations into when proteins are required and their dynamics. A large selection of antibodies or epitope-tagged transgenes are publicly available for the cytological examination of all meiotic stages. 

The organization of the Drosophila ovary facilitates cytological studies because it allows for the isolation, observation and staging of oocytes. The ovary contains several ovarioles, which are strings of developing oocytes arranged in temporal order, with stem cells at one end, and mature oocytes at the other. Each ovariole is divided into two main sections. The first, the germarium, contains the stem cells and early meiotic prophase; the stages which are present in the fetal oocytes of mammals. The second, the vitellarium, contains oocytes arrested in late meiotic prophase and is mostly concerned with development and growth of the oocyte. At the end of the vitellarium, meiosis resumes with nuclear envelope breakdown and entry into prometaphase I. The mature oocyte arrests in metaphase I until ovulation occurs. 

## 2. The Mitotic Region (Region 1)

At one end of the ovariole is region 1, which includes the germline stem cells (Figure 1). The presence of stem cells in the ovary is a significant difference compared to mammalian female meiosis, where a stem cell population is absent in the adult. An asymmetric division generates another stem cell and a cystoblast. This cystoblast then undergoes four incomplete mitotic divisions to generate a 16-cell cyst. The incomplete divisions result in all 16-cells of a cyst being connected by channels or “ring canals” [5]. 

It is unclear if events in region 1 and the mitotic cyst divisions are important for initiating meiosis. Meiotic cohesins function in these cells [6], homologous chromosome become paired [7] and synaptonemal complex (SC) proteins can be detected on the chromosomes [8,9]. As in most other model organisms, the signals controlling entry into female meiosis are poorly understood [10,11,12]. An important regulatory step is the concentration of transcripts into the oocytes, which depends on RNA binding proteins such as ORB [13] and Egalitarian [14] and interactions with BicD and the microtubules [15]. While two of the cells in the 16-cell cysts have four ring canals and simultaneously enter meiosis [16,17], one of these cells accumulates many transcripts, including some required for meiosis [18]. Concentrating these factors into the oocyte is required to maintain meiosis and specify the oocyte [19].

## 3. Meiotic Prophase (Regions 2–3)

Two cells in each cyst (“pro-oocytes) undergo premeiotic DNA replication [20] and then assemble the synaptonemal complex (SC), initiate double strand breaks (DSBs), and repair some of these as crossovers. This all occurs before the end of the germarium (Figure 1). The cytological aspects of meiotic progression were first described using electron microscopy along with an analysis of the first mutants to affect recombination nodule structure [16,21]. The SC initiates first at the centromeres [22,23] and forms without DSBs and possibly before them [24], which is a little unusual but not unique [25]. This conclusion was initially made by examining SC in recombination-defective mutants, but was later confirmed using an antibody to the phosphorylated variant of H2A (γH2AV) [26,27]. Conserved elements of SC assembly have been described, including the transverse protein C(3)G, other associated SC proteins [28,29,30] and two complexes of meiosis-specific cohesins [6,31,32]. Like other organisms, DSBs depend on the SPO11/TOPVIA orthologue MEI-W68 [33] and its TOPVIB partner MEI-P22 [34,35].

*Drosophila* has proved to be an excellent system to study the genetic controls on the distribution of crossing over [36,37]. Genes required for crossing over were originally classified as *precondition* or *exchange* genes, based on whether the mutants altered the distribution of crossing over [38]. Based on these genetic phenomena, it was proposed that precondition genes have a role in determining the location of crossover sites, while exchange genes execute the process. This proposal has turned out to be essentially correct [39,40,41]. The precondition genes include a complex of Minichromosome Maintenance (MCM) protein paralogs, the “mei-MCM” complex [42]. The exchange genes encode proteins such as XPF ortholog MEI-9 and MUS312 that are in the pathway that generates crossovers [43]. Similar to studies in other organisms, however, the generation of crossovers can occur by multiple pathways and involve antagonism with Blm helicase [41,42].

Studying chromosome rearrangements has revealed three structural features of meiotic recombination. First, the centromere effect of suppressing crossing over around the centromeric regions was discovered in *Drosophila* [44,45]. Second, the interchromosomal effect was the discovery that suppression of crossing over on one chromosome results in increased crossing over on other chromosomes [46]. This increase derives from a redistribution of crossovers among existing DSBs [47]. It has been proposed that the increased crossing over is the product of a checkpoint mechanism that detects rearrangement heterozygosity and causes a prolonged phase during which crossover sites can be established [48,49]. Third, the heterozygosity of chromosome rearrangement breakpoints suppresses crossing over at long distances. These observations have been interpreted as evidence for specialized sites that promote and regulate homologous chromosome interactions [50]. Pairing of homologous chromosomes is not disrupted in translocation [51] or inversion [52] heterozygotes, leading to the suggestion that meiotic crossing over depends on continuity of meiotic chromosome structure within large chromosomal domains. 

## 4. The Long Pause (Stages 2–12)

Similarly to mammals, there is a long pause in meiotic prophase, between pachytene when recombination occurs and metaphase I when a meiotic spindle assembles. As the cyst leaves the germarium, the oocyte is selected and the other pro-oocyte becomes a nurse cell (Figure 1). The oocyte loses its SC, although some SC components remain associated with the centromeres [53]. During this time, the oocyte grows and acquires cytoplasm from the nurse cells. By stage 13, the entire cytoplasmic contents of the nurse cells are transported, or “dumped” into the oocyte (Figure 2). 

## 5. Entry into the Meiotic Divisions (Stage 13–14)

Oocyte spindle assembly is acentrosomal in *Drosophila*, as in other animals, and was one of the first systems where this was studied genetically and cytologically. An important concept is that microtubules originating cytologically or cortically are recruited by the chromosomes [54]. Although centrioles lack [55] spindle pole components, including γ-tubulin, MSPS/XMAP215 and TACC are present and important for spindle organization [56,57,58]. 

In the absence of centrosomes, the chromosomes recruit microtubules (it is not known if they nucleate their growth). Given the size of the oocyte (400 µm × 100 µm), the mass of chromosomes (or “karyosome”) (5 µm) and the spindle (10 µm long) is quite small. Therefore, it is important for the oocyte to promote microtubule assembly around the chromosomes while inhibiting it in the rest of the cytoplasm. Indeed, the cytoplasm is rich in microtubules [59,60] and it is logical that the chromosomes have a primary role in controlling and initiating spindle assembly. 

Two mechanisms promote spindle assembly around the oocyte chromosomes. First, microtubule-associated proteins in the cytoplasm are inhibited, at least in part by the 14-3-3 protein [61]. Second, the localization of the Chromosome Passenger complex (CPC) to the chromatin promotes spindle assembly [62,63,64], with a contribution from the Ran pathway [65]. Among the proteins activated by the CPC are multiple kinesin proteins that bundle microtubules. The Kinesin 6 Subito, which most likely interacts with the CPC, recruits γ-tubulin to nucleate microtubule assembly [66]. In addition, the CPC is required for the assembly of the kinetochore, which directly interacts with microtubules at the centromeres [67]. The kinetochore orchestrates meiosis-specific functions such as fusion of sister centromeres [68] and bi-orientation of homologous chromosomes (see below). How tension works in this process is not understood, and it is surprising that in the absence of tension, metaphase I arrest is bypassed, resulting in precocious anaphase (Figure 2) [69,70].

Centrosomes not only nucleate and recruit microtubules, as they also help to organize them into a bipolar spindle (Figure 2). Data from studies on the kinesin-6 Subito have shown this is an active process, rather than an intrinsic property of microtubules. Subito organizes a central spindle, which contains several proteins [71]. Subito recruits the CPC to the central spindle, and in the absence of these proteins, bi-orientation is defective [72,73]. Interactions between the central spindle and the actin network also appear to be important for bi-orientation [74]. An interesting possibility is that interactions between kinetochores and central spindle-associated CPC are required for the process of error correction [63,75]. Indeed, the presence of a robust central spindle may help to compensate for the absence of centrosomes. 

Unlike mammals, the oocytes do not age as the mother does. This is because germ line stem cells are present in the adult female and produce a constant supply of new oocytes. Similarly to mammalian oocytes, however, there is a relatively long pause between prophase and metaphase. During this time, centromere and cohesion proteins must be maintained [76,77]. Meiotic cohesion is protected by PP2A [78] and the inhibition of Polo kinase [68,79]. By manipulating culture conditions and forcing females to hold their oocytes, it has been observed that aging mature oocytes causes an increase in chromosome segregation errors [77,80] and defects in meiotic spindle organization [81]. It has been proposed these defects are due to the levels of superoxide dismutase [82,83] or decreases in translation [81]. 

While chiasmata direct the segregation of most chromosomes, *Drosophila* females have a second system for chromosome segregation. Known as the achiasmate system, it is required for the segregation of the naturally achiasmate 4th chromosomes, as well as any other chromosomes that are achiasmate either by chance (so-called E0s) or because of heterozygosity for a balancer chromosome [84,85]. Although some of the genetics of achiasmate segregation is esoteric, this system clearly depends on microtubule-associated proteins. For example, NOD is a conserved kinesin 10 and one of the first proteins required for the achiasmate system to be identified [86]. Nonetheless, the role of the achiasmate system in normal Drosophila females has not been determined. Is it only required for the 4th chromosome and rare achiasmate chromosomes, or is it more intricately involved in the segregation of all chromosomes? 

## 6. Exit from Meiosis and Fertilization (the Embryo)

The events that occur between metaphase I and fertilization are rapid. The oocyte is activated by their passage down the oviduct, independent of fertilization, and rapidly completes the two meiotic divisions [87,88,89]. *Drosophila* oocytes do not excise polar bodies. Instead, all four meiotic products align perpendicularly to the oocyte cortex [90,91]. The inner-most meiotic product fuses with the male pro-nucleus, while the remaining three female meiosis products fuse and form a single polar body that can persist for a long time [92].

There are interesting consequences of producing polar bodies in meiosis. For example, inversion heterozygotes may not have fertility defects because the crossover products, which generate chromosome rearrangements, end up in the polar bodies. The fact that only one meiotic product from female meiosis is fertilized has led to theories that there is competition between chromosomes, based on centromere strength, for inclusion in the zygote [93]. Unequal centromeres could interact with an asymmetric spindle to result in the biased retention of certain chromosomes in the zygote [94,95]. Genetic studies in Drosophila have generated evidence of asymmetric segregation that was influenced by “centromere strength” [96,97,98,99]. More recently, there is cytological evidence of an asymmetric spindle [100] and genetic data suggesting a centromere drive [101].

## 7. Summary

Drosophila has been a model system since the discovery of nondisjunction [102,103] the first meiotic mutant [104] and the first screens for meiotic mutants [105,106]. Now, with a variety of genetic and cytological tools as described above, Drosophila remains a model organism of choice for the study of meiosis.

## Figures and Tables

**Figure 1 genes-13-00775-f001:**
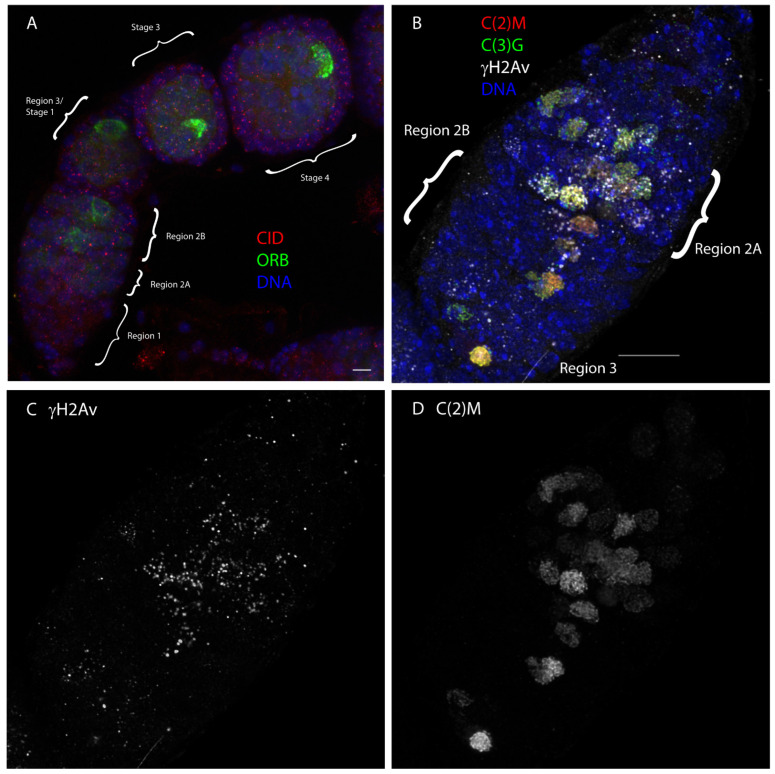
The early stages—meiotic prophase. (**A**) Germ cell development in the adult begins with stem cells located in the anterior tip of the germarium located in region 1. Four mitotic division in region 1 produce a 16-cell cyst, marked by the ORB protein in green. Surrounding the 16 cells cysts are somatic follicle cells. One of the two cells with four ring canals becomes specified as the oocyte and accumulates the most ORB protein, as well as many other RNAs and proteins, including those required for meiosis. The image also includes centromere staining (CID) in red, DNA in blue, and the scale bar is 10 um. (**B**–**D**) Germarium showing appearance of SC and double strand breaks. The cohesin C(2)M is in red, SC transverse filament protein C(3)G in green and double strand break marker γH2AV in white. SC assembly initiates in region 2A, double strand breaks are observed shortly after, and by region 3 the DNA is repaired and crossovers have formed. The SC dissolves around stage 4 or 5.

**Figure 2 genes-13-00775-f002:**
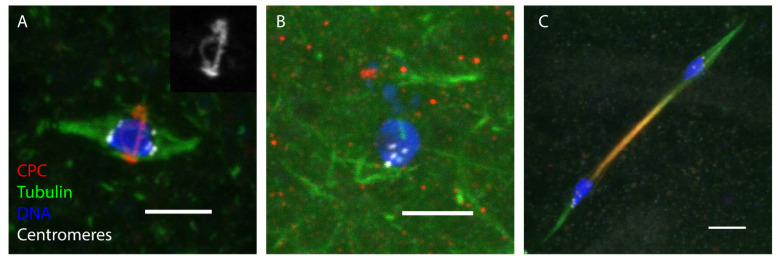
The latest stage, meiotic metaphase. In stages 13–14, the nuclear envelope breaks down and the meiotic spindle assembles. Meiosis arrests in metaphase I and progresses into anaphase I and meiosis II only when the oocyte passes through the oviduct on the way to being fertilized. In all images, the spindle is in green, the chromosome passenger complex (CPC) in red, centromeres in white, DNA in blue, and the scale bar is 5 µm. (**A**) A metaphase I spindle in a stage 14 oocyte. The inset shows that the central spindle proteins are organized in a microtubule-associated ring that goes around the chromosomes and is perpendicular to the spindle. (**B**) The CPC is required for spindle assembly. An oocyte expressing an shRNA for RNAi against INCENP causes the loss of all spindle microtubules. (**C**) In the absence of chiasma and tension, precocious anaphase is observed. This is an oocyte homozygous for a *mei-P22* mutation, which eliminates all meiotic recombination.

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
