# Peer review of "A Brief History of *Drosophila* (Female) Meiosis"

_genes, 2022, doi:10.3390/genes13050775_

Round 1

Reviewer 1 Report

This review of Drosophila meiosis is written as a brief introduction to scientists that are new to the field. I appreciate a need for this type of brief introduction, but there is already a very accessible introduction to Drosophila meiosis written by Hughes et al 2018.

My major concern with this manuscript is that it isn’t polished enough. First, there are numerous vocabulary errors that suggest the wrong meaning. For example, on line 42, the authors refer to “asynchronous division” of the stem cell, but I’m certain they meant “asymmetric division”. Additionally, on line 62, the authors describe how transcripts are concentrated into oocytes “which depends on RNA binding proteins such as ORD…”. I’m certain they mean “orb” as “ord” is a sister chromatin cohesin protein. While these could be viewed as typos, I view them as serious mistakes that take away from the credibility of the manuscript. Secondly, the phrasing and word choice is not scientifically meaningful or accurate in places. For example, on lines 110 and 111, the authors say “there is a long pause in meiotic prophase, between the recombination phase and the spindle phase.” I can’t place any scientific meaning in those terms, particularly “spindle phase.” Finally, there are places throughout the manuscript where the authors are trying to provide field-specific information while also not overwhelming the reader with too much information. However, this frequently causes vague writing that would confuse me if I was new to the field. As an example, on line 117, the authors say that “By stage 13, the nurse cells have dumped their entire contents and only the oocyte remains.” As an expert, I know that the function of nurse cells is to provide support for the oocyte and that they do this by ultimately “dumping” their cytoplasmic contents into the oocyte. However, the authors state that their audience is people who are new to the field, and so this idea of nurse cells dumping their contents is too vague without additional explanation and only leads to confusion. There are instances of this throughout the manuscript.

As one minor concern, there are many instances where a reference is listed as “Error! Reference source not found.” I cannot thoroughly evaluate a review manuscript if I can’t check the references.

Author Response

My major concern with this manuscript is that it isn’t polished enough.

We have made several editorial changes to try and improve the manuscript.

For example, on line 42, the authors refer to “asynchronous division” of the stem cell, but I’m certain they meant “asymmetric division”.

Changed to “asymmetric”

Additionally, on line 62, the authors describe how transcripts are concentrated into oocytes “which depends on RNA binding proteins such as ORD…”. I’m certain they mean “orb” as “ord” is a sister chromatin cohesin protein.

Changed to “ORB”

the phrasing and word choice is not scientifically meaningful or accurate in places. For example, on lines 110 and 111, the authors say “there is a long pause in meiotic prophase, between the recombination phase and the spindle phase.” I can’t place any scientific meaning in those terms, particularly “spindle phase.”

Removed term “spindle phase”. Read through manuscript and changed some other terminology.

Finally, there are places throughout the manuscript where the authors are trying to provide field-specific information while also not overwhelming the reader with too much information. However, this frequently causes vague writing that would confuse me if I was new to the field. As an example, on line 117, the authors say that “By stage 13, the nurse cells have dumped their entire contents and only the oocyte remains.” As an expert, I know that the function of nurse cells is to provide support for the oocyte and that they do this by ultimately “dumping” their cytoplasmic contents into the oocyte. However, the authors state that their audience is people who are new to the field, and so this idea of nurse cells dumping their contents is too vague without additional explanation and only leads to confusion.

Added description of what is meant by “dumping”

As one minor concern, there are many instances where a reference is listed as “Error! Reference source not found.” I cannot thoroughly evaluate a review manuscript if I can’t check the references.

This was due to an error caused when the manuscript was uploaded and converted to a Genes format. This has been corrected by converting the Endnote citations to plain text.

Reviewer 2 Report

In this review “A brief history of Drosophila meiosis” the authors provide an overview of the female meiosis in Drosophila melanogaster with an interesting focus on the first observations pointing out the pioneering works done in the ’70-’80 via EM.

Although the review is very interesting, I found it a little bit difficult to read as some part are repetitive.

I also have some comments:

-the title should specify that the focus is on female meiosis because, as the authors stated in the Introduction, male meiosis il a little bit different

- The abstract is too short; I think there is space to describe the structure of this review. Again, this is a good place to clarify that it is about female meiosis.

-some references are missing and highlighted by the journal with “Error! Reference source not found”

-line 22: RNAi is normally used for knock-down, then I would suggest to add “…. Ca be used to knock out or knock down…”

-lines 56-57. “It is unclear when meiosis initiates….” I think this sentence need to be explained with more details.

-line 58: SC= synaptonemal complex. Abbreviations should be explained the first time that are used while the authors do it only at line 71

-line 62: the protein is Orb non ORD. Moreover, Drosophila proteins are not all capital.

-line 65: simultaneously enter meiosis. Reference?

-line 79: “associated other SC proteins” do the authors mean “other associated SC proteins”?

-lines 83-85. The sentence about DSBs staining is not necessary

-Lines 97-99 are repetitive sentences.

-Figure Legend 2: the chromosomal passenger complex (CPC). Abbreviations should be explained at the beginning, not in panel B.

-line 134: do the authors mean (gamma) γ-tubulin? Or Msps? Because tubulin is generic

-line 137: “the chromosomes recruit microtubules” Do they authors mean “nucleate” instead of “recruit”?

-line 141: the cytoplasm is rich in microtubules or tubulin?

-lines 158-159. “In the absence of centrosomes, the microtubules are organized into a bipolar spindle without pre-existing spindle pole”. This sentence is not clear: what do they mean for pre-existing spindle poles? And again, this has been already explained at the beginning of the chapter (Lines 130-onward)

-line 183: Nod protein with capital N

-line 187: Is it a question? Is a “?” missing at the end?

Author Response

Although the review is very interesting, I found it a little bit difficult to read as some part are repetitive.

We worked on reorganizing certain parts to avoid repetition.

-the title should specify that the focus is on female meiosis because, as the authors stated in the Introduction, male meiosis il a little bit different

Added the word “female’ to the title

- The abstract is too short; I think there is space to describe the structure of this review. Again, this is a good place to clarify that it is about female meiosis.

Added text to the Abstract summarizing what the article is about and why males are not part of the review.

-some references are missing and highlighted by the journal with “Error! Reference source not found”

As noted to Reviewer 1, this was due to an error caused when the uploaded manuscript was converted to a Genes format. This was corrected by converting the reference fields to plain text.

-line 22: RNAi is normally used for knock-down, then I would suggest to add “…. Ca be used to knock out or knock down…”

Changed to “knock out or knock down”

-lines 56-57. “It is unclear when meiosis initiates….” I think this sentence need to be explained with more details.

Changed the wording of this sentence with details to imply if the events occurring in mitotic cells as described in the previous paragraph are important for meiosis. Examples are listed in the next sentence.

-line 58: SC= synaptonemal complex. Abbreviations should be explained the first time that are used while the authors do it only at line 71

SC explained earlier

-line 62: the protein is Orb non ORD. Moreover, Drosophila proteins are not all capital.

Changed to ORB. Using the convention that abbreviations are all caps. 

-line 65: simultaneously enter meiosis. Reference?

Added reference

-line 79: “associated other SC proteins” do the authors mean “other associated SC proteins”?

Yes - corrected

-lines 83-85. The sentence about DSBs staining is not necessary

Modified this sentence but kept it because we think it is important to indicate how (and the only way) to detect DSBs in Drosophila.

-Lines 97-99 are repetitive sentences.

Moved this sentence up to avoid repetition.

-Figure Legend 2: the chromosomal passenger complex (CPC). Abbreviations should be explained at the beginning, not in panel B.

corrected

-line 134: do the authors mean (gamma) γ-tubulin? Or Msps? Because tubulin is generic

Gamma tubulin.  The symbol comes a across a little small. Added Msps and Tacc.

-line 137: “the chromosomes recruit microtubules” Do they authors mean “nucleate” instead of “recruit”?

We meant recruit because it is not clear the microtubules grow from the chromosomes.

-line 141: the cytoplasm is rich in microtubules or tubulin?

Both, but we mean to emphasize microtubules.

-lines 158-159. “In the absence of centrosomes, the microtubules are organized into a bipolar spindle without pre-existing spindle pole”. This sentence is not clear: what do they mean for pre-existing spindle poles? And again, this has been already explained at the beginning of the chapter (Lines 130-onward)

Changed the sentence to emphasize the role of centrosomes in organizing a bipolar spindle.

-line 183: Nod protein with capital N

Corrected

-line 187: Is it a question? Is a “?” missing at the end?

Question mark added.